# Image Retrieval Method Based on Visual Map Pre-Sampling Construction in Indoor Positioning

Jianan Bai [1], Danyang Qin [1,2,*], Ping Zheng [1] and Lin Ma [3]

1   Department of Electronic Engineering, Heilongjiang University, Harbin 150080, China
2   National Mobile Communications Research Laboratory, Southeast University, Nanjing 211189, China
3   Department of Electronics and Information Engineering, Harbin Institute of Technology, Harbin 150080, China
*   Correspondence: qindanyang@hlju.edu.cn

**Abstract:** In visual indoor positioning systems, the method of constructing a visual map by point-by-point sampling is widely used due to its characteristics of clear static images and simple coordinate calculation. However, too small a sampling interval will cause image redundancy, while too large a sampling interval will lead to the absence of any scene images, which will result in worse positioning efficiency and inferior positioning accuracy. As a result, this paper proposed a visual map construction method based on pre-sampled image features matching, according to the epipolar geometry of adjacent position images, to determine the optimal sampling spacing within the constraints and effectively control the database size while ensuring the integrity of the image information. In addition, in order to realize the rapid retrieval of the visual map and reduce the positioning error caused by the time overhead, an image retrieval method based on deep hashing was also designed in this paper. This method used a convolutional neural network to extract image features to construct the semantic similarity structure to guide the generation of hash code. Based on the log-cosh function, this paper proposed a loss function whose function curve was smooth and not affected by outliers, and then integrated it into the deep network to optimize parameters, for fast and accurate image retrieval. Experiments on the FLICKR25K dataset and the visual map proved that the method proposed in this paper could achieve sub-second image retrieval with guaranteed accuracy, thereby demonstrating its promising performance.

**Keywords:** indoor positioning; binary codes; semantics; image retrieval

## 1. Introduction

With the growth of the Internet and the popularity of mobile communication devices, location information has steadily evolved into the most significant information in people's daily lives, producing a slew of location-based services. However, people spend 80% of their time indoors [1]. Therefore, there is an urgent need for a convenient and fast method to achieve high-precision indoor positioning. In recent years, numerous indoor localization methods have been proposed: Bluetooth [2,3], Wi-Fi [4,5], infrared [6], ultra-wideband (UWB) [7], etc. Among them, visual indoor positioning, with its characteristics of low cost, convenient data acquisition, and significant adaptability, has developed into one of the mainstream indoor positioning technologies. Image-based indoor positioning uses the results of image retrieval directly as the positioning results [8]. Although the operation is simple, the positioning results are utterly dependent on the established offline database and image search, resulting in low accuracy. Geometric relationship-based visual indoor localization methods add a step to calculating the relative positional relationship between the query camera and the database camera, hence having higher localization accuracy [9,10]. In addition to using traditional vision sensors to capture image information, there are many ways to use depth cameras to obtain information about the surrounding environment. In [11,12], RGB-D cameras were used to capture RGB and depth images from

the surrounding environment to obtain camera poses and 3D coordinates of pixels. In order to improve positioning accuracy, some scholars have proposed a visual indoor positioning method based on multi-sensor or fusion information [13,14]. Although the above method has a high positioning accuracy, it requires being equipped with additional equipment to assist the visual sensor, which weakens the low deployment cost and strong adaptability of visual indoor positioning.

For existing visual indoor localization methods, it is typically essential to collect images and build an offline database, which is then used as the foundation for localization. A visual map, an image database supporting visual indoor positioning, stores the image information from the indoor environment as well as the position coordinates of each image, all of which are collected and stored in the offline stage of the system. Therefore, it is crucial to establish a visual map database with comprehensive image information and precise geographical coordinates. Consequently, this paper proposes a visual map construction method based on pre-sampled image similarity.

Following the creation of an offline image database, an efficient image retrieval method is required in order to acquire database images that are visually comparable to the query image. Generally, traditional feature extraction methods or convolutional neural networks are used to extract feature vectors from images, and local feature aggregation descriptors (VLAD) [15], the Fisher Vector, and BoW [9,10] are used to describe vectors to achieve image retrieval. To some degree, the preceding approaches may meet the requirements of indoor positioning. However, when the indoor area is large and the scene information is abundant, the scale of the visual map produced will become quite large. Accordingly, additional retrieval time will be consumed. More crucially, when it comes to moving-target positioning and navigation, the longer the image retrieval time, the farther the target moves, which in turn leads to larger positioning errors. To overcome this issue, an image retrieval method based on the deep hashing method was used in this paper. This method can map high-dimensional image features into a two-dimensional space and create hash codes, which greatly reduces the number of feature dimensions. In addition, since the hash codes are composed of a simple arrangement of 0 and 1, only a simple Hamming distance calculation is required to return results. This method will, therefore, take less retrieval time than other similarity metrics such as the complex Euclidean distance, Manhattan distance, and Mahalanobis distance.

The main contributions of this paper are summarized in Figure 1.

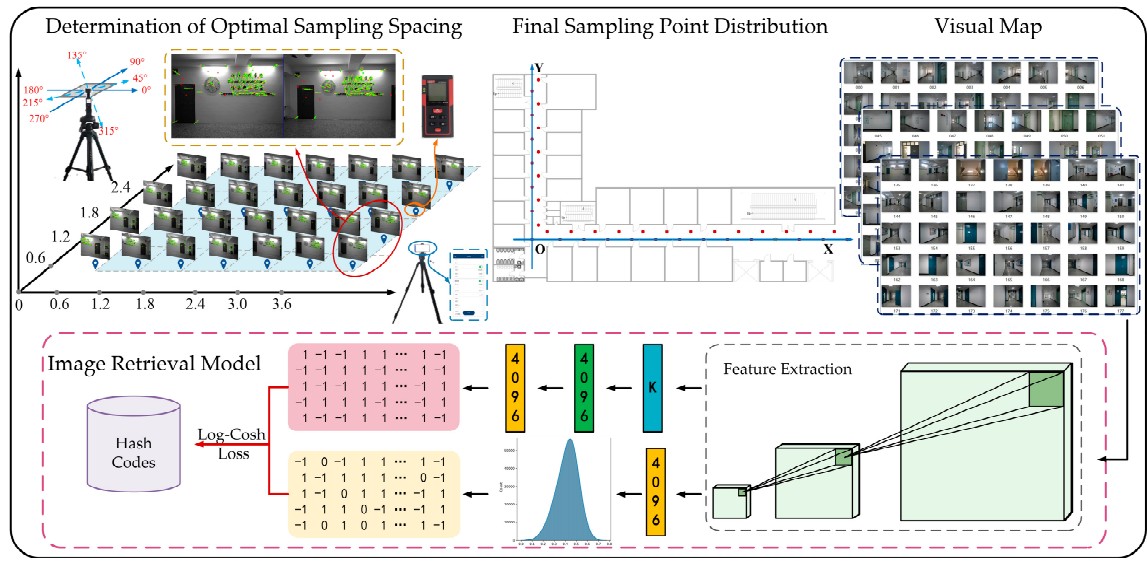

**Figure 1.** Roadmap of the research.

(1) A visual map construction method is proposed, based on pre-sampled image features matching, to solve the problem of redundant image samples and missing scenes features resulting in the erroneous setup of fixed sampling spacing in a varied indoor environment. While effectively controlling the scale of the database, the integrity of the visual features of indoor scenes is nevertheless maintained.

(2) A novel image retrieval model is established based on deep hashing to achieve high precision and speed in image retrieval on the constructed visual map of actual buildings, which has certain practical engineering significance.

## 2. Related Work

### 2.1. Visual Map

The methods of building a visual map can be summarized into two categories according to the sampling method, which are the video-stream-based construction method [16,17] and the point-by-point sampling method. The video-stream-based method has the advantage of simple sampling, but the non-uniform motion of the visual sensor can easily cause image redundancy or loss in some areas. Furthermore, the approach of directly using successive frames cannot directly query the geographic information of a specific location, and can only speculate from the previous frame, which has accumulated errors and certain limitations. In order to reduce the size of the database, some studies select frames in the video stream in a certain way and use them as keyframes to build a visual map database. In the literature [18], the keyframes are determined using the matching rate as the metric of choice to generate an offline image database. However, due to the movement speed of the vision sensor not being stable, it is difficult to accurately calculate the geographic coordinates of a specific position or keyframe, making it difficult to perform indoor positioning effectively. Therefore, this paper adopts a static point-by-point sampling method to construct a visual map. Using visual sensors to capture images at locations pre-marked in the indoor environment and recording the geographic coordinates of the locations avoids the problems of blurred images and uneven image distribution in some scenes caused by the dynamic acquisition method.

### 2.2. Content-Based Image Retrieval

Before performing the search, the content-based image retrieval system extracts features from the database images and describes the images using feature vectors so that there is a one-to-one correspondence between the images and the feature vectors, and finally returns the retrieved results by calculating and sorting the distance between the feature vectors. Therefore, retrieval performance is closely related to feature extraction, feature selection, and similarity measurement of matched images [19]. Among these, feature extraction and selection are the most influential factors in representing images' semantic content. These features can be divided into global features (colour features, texture features, shape features, etc.) that describe the entire image and local features (edges, corners, lines, etc.) that describe part of the image. When using the aforementioned low-level features to retrieve images, the practice of increasing the feature dimension is generally adopted to improve the accuracy of the retrieval results, so the retrieval time will also grow linearly with the increase in scene images. However, when it comes to the positioning and navigation of moving objects, a long retrieval time is bound to reduce the performance and efficiency of positioning. Therefore, it is necessary to choose a fast and accurate image retrieval method. In addition, these low-level features cannot comprehend the abundant semantic information in the image and cannot sufficiently express the user's purpose. With the continuous development of computer technology, deep-learning algorithms have been widely employed in the field of computer vision, and semantic-based image retrieval methods have also demonstrated improved performance.

Deep hashing can map high-dimensional image features into two-dimensional space, considerably reducing the dimension of features, and binary encoding has the advantage of rapid search. When mapping, a semantic similarity structure will be constructed to

guide the binary encoding based on image similarity to preserve the semantic relationship between different data points. Therefore, in order to adapt to today's trend of high-dimensional image features, to meet the user's need to search for desired information in the database, and to take into account the convenience of the user and the characteristics of the input image, this paper will use a deep hashing method to achieve the learning of image semantic information and efficient image retrieval. Inspired by [20] and based on the significant impact of data distribution on data-dependent hash learning, this paper also empirically investigates the deep feature statistics and designs a deep hash image retrieval method based on similarity reconstruction.

## 3. Methodology

### 3.1. Establishment of Indoor Positioning Image Database

Visual indoor localization is a special application of content-based image retrieval (CBIR). Image retrieval returns the database picture with comparable scene information to the query image, and the geographical coordinates of the query image may then be determined based on the annotated geographical coordinates of the database image. Therefore, the results of localization are extremely dependent on the image quality and the accuracy of the geographical coordinates of the offline database. The video-based approach to building a visual map has the advantage of simple sampling, but the visual sensor motion is not stable, which tends to lead to image blurring and to reduce the efficiency and the results of image matching. In addition, vision sensors struggle to maintain uniform motion, and non-uniform motion complicates the computation of geographical coordinates, resulting in incorrect location findings. Hence, in order to eliminate image blurring and ensure the reliability of geographical coordinates, this paper used a static image acquisition method. The equipment used to construct the visual map is shown in Figure 2. First, the image was captured using a DJI Pocket2 with three-axis mechanical stabilization, and the resolution of the image was 4608 × 3456. Then, the coordinates of the position of the equipment in the world coordinate system were calculated based on the laser rangefinder, and the angle of the capture was recorded using MATLAB Mobile. A mobile phone can be connected to MATLAB using this app, and the direction of the device can be collected by using the sensor module, including the azimuth, pitch, and roll. Additionally, sensor data can be streamed directly to the MathWorks Cloud for analysis, and data can be saved offline. For collecting complete picture information surrounding the shooting point, the camera captured pictures from eight different angles, as illustrated in Figure 3. Considering that the natural lighting in the indoor environment is not steady, artificial lighting was used to reduce blur and keep image noise to a minimum in order to decrease the influence of lighting variations on image retrieval.

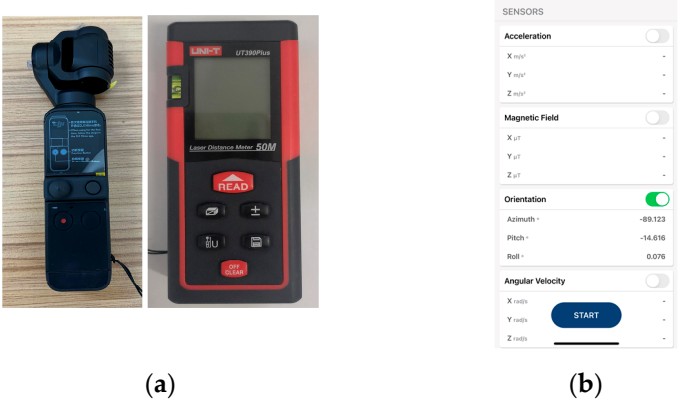

**(a)** **(b)**

**Figure 2.** (**a**) DJI Pocket2 and laser rangefinder; (**b**) MATLAB Mobile.

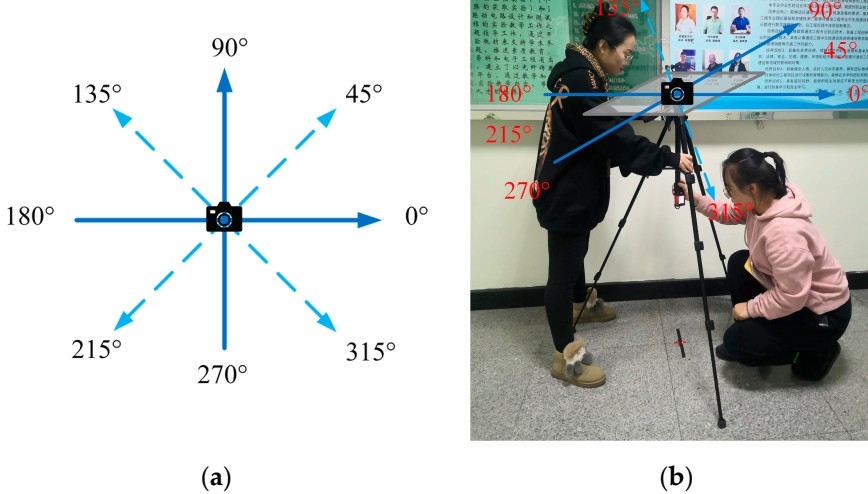

(**a**)  (**b**)

**Figure 3.** (**a**) Image capture angle; (**b**) determination of image acquisition position.

Using the point-by-point sampling approach, however, a small sample distance results in redundant picture information and a greater database size, which increases the retrieval complexity and time consumption. Excessive sampling spacing will lead to the lack of image information and coordinate information for some scenes in the visual map, limiting the accuracy of image retrieval or even making it impossible, resulting in the inability to achieve positioning in the online stage. As a result, and based on epipolar geometry, this paper proposes a visual map construction method that uses feature point matching, so as to determine the optimal sampling interval within the constraints and effectively control the scale of the database. As shown in Figure 4, the relative positional relationship between the query camera and the database camera can be estimated according to the epipolar geometry.

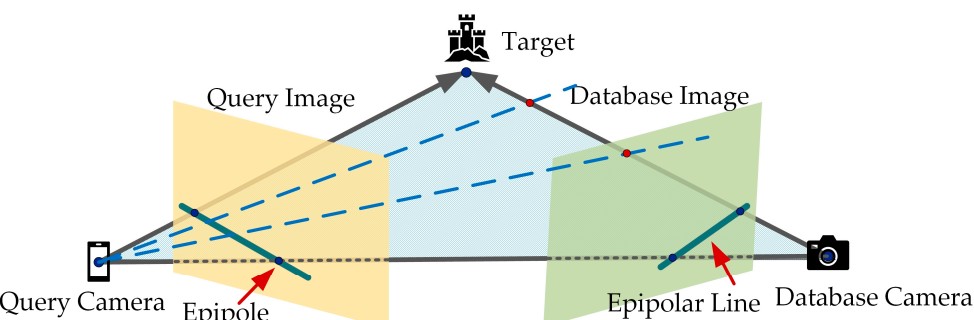

**Figure 4.** Epipolar geometry.

According to the characteristics of the epipolar geometry, the electronics laboratory building of Heilongjiang University was used as the experimental scene, and images were captured every 0.6 m. In order to obtain the optimal sampling spacing in the vertical axis direction, an image pair consisting of a reference image with the same horizontal coordinate and a pre-sampled image was tested, and the results obtained are shown in Table 1. Here, the green points are inliers, the red points are outliers, and the yellow straight line represents the relative direction between the two images, which is consistent with the relative direction between the sampling points in the real scene. The next steps were to observe the match of feature points between the reference image and the images acquired at different coordinates; define the coverage ratio $C_d$ as the ratio of feature points between the sampled image and the corresponding reference image; and define the match ratio $M_d$ as the ratio of the number of matched points in the image to the total number of feature points.

The specific form of the coverage ratio and the match ratio is shown in Formulas (1) and (2).

$$C_d = \frac{N_{sam}}{N_{cor}} \tag{1}$$

$$M_d = \frac{N_{match}}{N_{total}} \tag{2}$$

**Table 1.** Epipole distribution of the images collected at each coordinate point.

| Baseline Image Coordinates (m) | Image Coordinates (m) | | Image Coordinates (m) | | Image Coordinates (m) | |
|---|---|---|---|---|---|---|
| (0, 0) | (0, 0.6) | Coverage Ratio 0.837 Match Ratio 0.753 | (0, 1.2) | Coverage Ratio 0.787 Match Ratio 0.746 | (0, 1.8) | Coverage Ratio 0.743 Match Ratio 0.677 |
| (0.6, 0) | (0.6, 0.6) | Coverage Ratio 0.775 Match Ratio 0.808 | (0.6, 1.2) | Coverage Ratio 0.678 Match Ratio 0.713 | (0.6, 1.8) | Coverage Ratio 0.681 Match Ratio 0.639 |
| (1.2, 0) | (1.2, 0.6) | Coverage Ratio 0.833 Match Ratio 0.750 | (1.2, 1.2) | Coverage Ratio 0.790 Match Ratio 0.697 | (1.2, 1.8) | Coverage Ratio 0.728 Match Ratio 0.543 |
| (1.8, 0) | (1.8, 0.6) | Coverage Ratio 0.761 Match Ratio 0.843 | (1.8, 1.2) | Coverage Ratio 0.716 Match Ratio 0.755 | (1.8, 1.8) | Coverage Ratio 0.689 Match Ratio 0.612 |
| (2.4, 0) | (2.4, 0.6) | Coverage Ratio 0.772 Match Ratio 0.765 | (2.4, 1.2) | Coverage Ratio 0.784 Match Ratio 0.591 | (2.4, 1.8) | Coverage Ratio 0.682 Match Ratio 0.473 |
| (3.0, 0) | (3.0, 0.6) | Coverage Ratio 0.839 Match Ratio 0.799 | (3.0, 1.2) | Coverage Ratio 0.828 Match Ratio0.662 | (3.0, 1.8) | Coverage Ratio 0.778 Match Ratio 0.493 |
| (3.6, 0) | (3.6, 0.6) | Coverage Ratio 0.839 Match Ratio 0.765 | (3.6, 1.2) | Coverage Ratio 0.805 Match Ratio 0.759 | (3.6, 1.8) | Coverage Ratio 0.756 Match Ratio 0.483 |

Statistical analysis was performed on the obtained experimental data, and as it was found that the coverage ratio was more likely to fall at or near 0.7, that value was selected. Due to the small error in solving the fundamental matrix when the feature point matching rate was above 50% in the online positioning stage, the matching rate was determined to be 0.5. Finally, based on the fact that the corridor width of the electronic laboratory building is 2 m and the physical structure of the building is tortuous and has a strong sense of depth, the optimal sampling spacing for the experimental environment was determined to be 3 m × 1.8 m. Figure 5 depicts the specific distribution of sample sites in the experimental situation, with the lower left corner representing the origin position and the red markers representing the sampling locations.

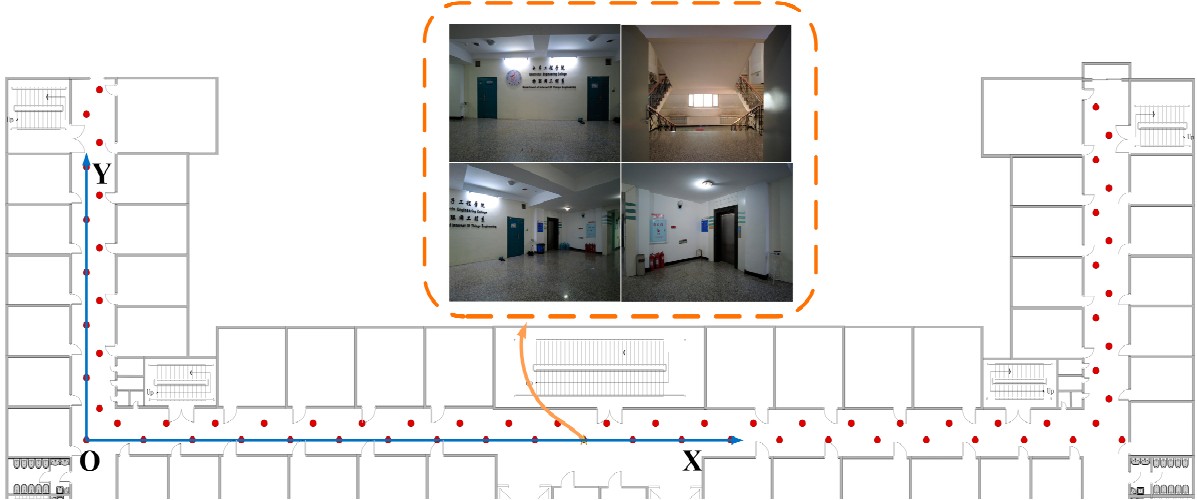

**Figure 5.** Distribution of sampling points in the electronics laboratory building.

### 3.2. Deep Hash Model

Figure 6 shows the overall process of image retrieval based on deep hashing.

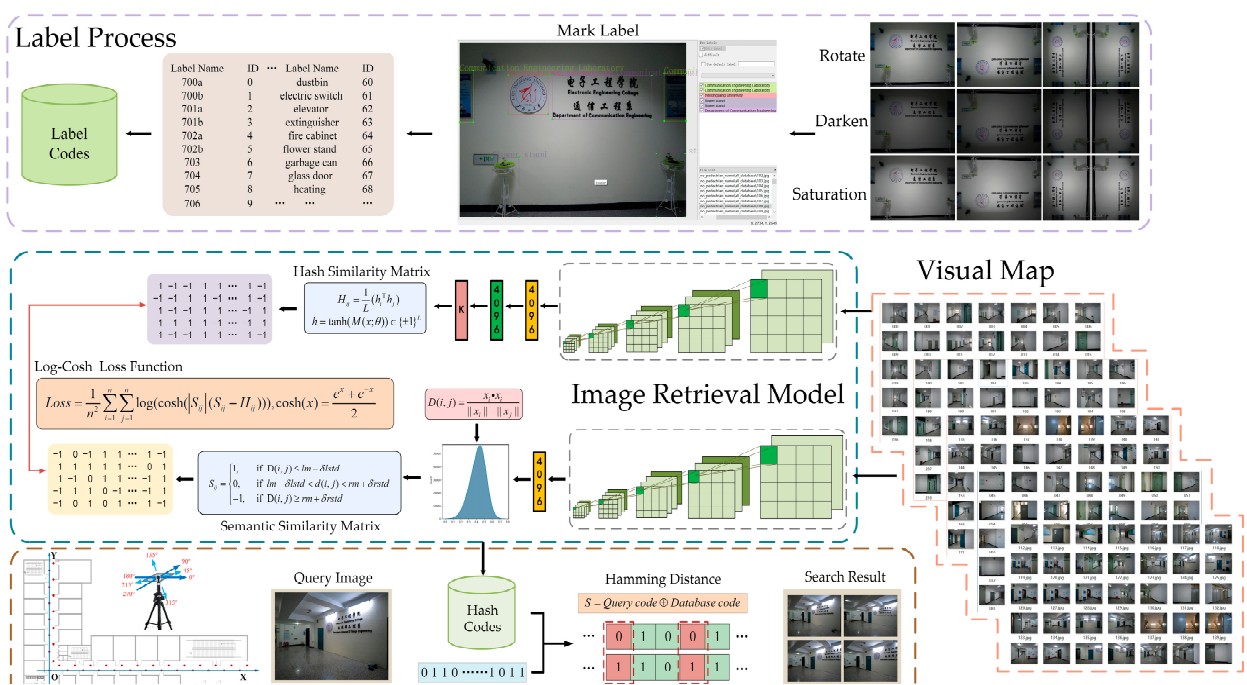

**Figure 6.** Flowchart of image retrieval based on deep hashing.

First, the database image is preprocessed to obtain the binary code of the label. Second, image retrieval based on deep hashing learns a feature mapping function through the similarity of images so that it can map the binary code according to the input image. Finally, the Hamming distance between the binary codes is calculated as the foundation for judging the similarity between the images. Suppose the dataset contains $n$ images, $X = \{x_i\}_{i=1}^{n}$ represents the dataset, and $x_i$ is the $i$th image in the dataset. The hash code of each image in the dataset is learned by the hashing method, and then the binary code of the query image is generated according to the constructed hash function $M(x_i;\theta)$, where $\theta$ represents the

parameters of the neural network. When mapping an image to a two-dimensional space, the $sign(x)$ function is used to perform a binarization operation, as shown in Formula (3).

$$sign(x) = \begin{cases} 1 & (x \geq 0) \\ -1 & (x < 0) \end{cases} \tag{3}$$

The obtained hash code can be represented by $B = \left\{ b_i \in \{\pm 1\}^L \right\}_{i=1}^n$, where $b_i$ represents the $i$th hash code and $L$ represents the length of the hash code, which can generally be set to 16 bits, 32 bits, 64 bits, and 128 bits. Semantic learning of images can be accomplished by constructing a semantic similarity matrix between image pairs. In this paper, first, images were randomly extracted from the dataset by a pre-trained VGG-F network, and the features from the fc-7 layer were extracted to construct the similarity matrix. Second, the similarity of each pair of data points was calculated for the extracted deep features considering the data-dependent hashing algorithm.

Image similarity is usually measured using Euclidean distance and cosine distance. Euclidean distance can reflect the absolute difference of individual numerical characteristics, while cosine distance is more about distinguishing differences in direction and is not sensitive to absolute values. Refer to [20] and calculate the cosine distances of the visual map images to generate the distribution displayed in Figure 7. Divide it into two parts according to the maximum value of the histogram, and use two semi-Gaussian distributions to fit them, respectively. The constructed semantic similarity matrix is as follows:

$$S_{ij} = \begin{cases} 1, & \text{if } D(i,j) \leq lm - \delta lstd \\ 0, & \text{if } lm - \delta lstd < d(i,j) < rm + \delta rstd \\ -1, & \text{if } D(i,j) \geq rm + \delta rstd \end{cases} \tag{4}$$

where $D(i,j)$ denotes the cosine distance between data point $x_i$ and data point $x_j$, $lm$ and $lstd$ are the mean and standard deviation of the Gaussian distribution on the left, $rm$ and $rstd$ are the mean and standard deviation on the right, and $\delta$ is the key parameter for constructing the semantic similarity matrix, which will continue to be discussed in the later experiments. $S_{ij}$ is set to 1 when the semantic similarity between $x_i$ and $x_j$ is high. On the contrary, $S_{ij}$ is set to $-1$ when $x_i$ and $x_j$ are not semantically similar. When the similarity cannot be judged, $S_{ij}$ is set to 0. The data points in the visual map may be mapped to hash codes using the principle that hash codes describe data points with similar semantics and dissimilar hash codes describe data points with distinct semantics.

However, the $sign$ function cannot be used for gradient training and should be replaced with a differentiable relaxation function for model training. Among the numerous alternative functions, the tanh function achieves the approximation and similarity of the $sign$ function and makes up for the disadvantage that the $sign$ function is difficult to optimize because of its discreteness. As a result, the tanh function is chosen for network training, and the hash code $h = \tanh(M(x;\theta)) \in \{\pm 1\}^L$ is obtained. The similarity matrix is obtained by calculating the inner product of the hash codes, which can be used to judge whether the binary hash codes are similar or not. The constructed similarity matrix is as follows:

$$S_{ij}^h = \frac{1}{L}(h_i^{\mathrm{T}} h_j) \tag{5}$$

In deep-hashing-based image retrieval algorithms, the mean squared error (L2 loss) is typically employed to improve the network so as to retain the semantic structure and minimize the difference between the semantic similarity matrix and the hash similarity matrix. This method has the advantage of being simple to calculate. However, the robustness of the algorithm is low, and it is easily affected by outliers, resulting in a loss that cannot be ignored. Consequently, the log-cosh loss, which is smoother than the mean squared error,

was utilized in this research. The specific expressions of the mean squared error and the log-cosh function are as follows:

$$MSE = \sum_{i=1}^{n} \left( y_i - y_i^p \right)^2 \tag{6}$$

$$L(y, y^p) = \sum_{i=1}^{n} \log\left( \cosh\left( y_i^p - y_i \right) \right) \tag{7}$$

where the cosh function is

$$\cosh(x) = \frac{e^x + e^{-x}}{2} \tag{8}$$

It can be observed that when the sample data $y$ is small, the log-cosh loss can be approximated to $\frac{y^2}{2}$, and when the sample data $y$ is large, the log-cosh loss can be approximated to $|y| - \log(2)$. From this, it can be inferred that the log-cosh loss has similar properties to the mean squared error but is more robust and less susceptible to outliers. As a result, the loss function designed in this paper based on the log-cosh function was as follows:

$$Loss = \frac{1}{n^2} \sum_{i=1}^{n} \sum_{j=1}^{n} \log(\cosh(|S_{ij}|(S_{ij} - S_{ij}^h))) \tag{9}$$

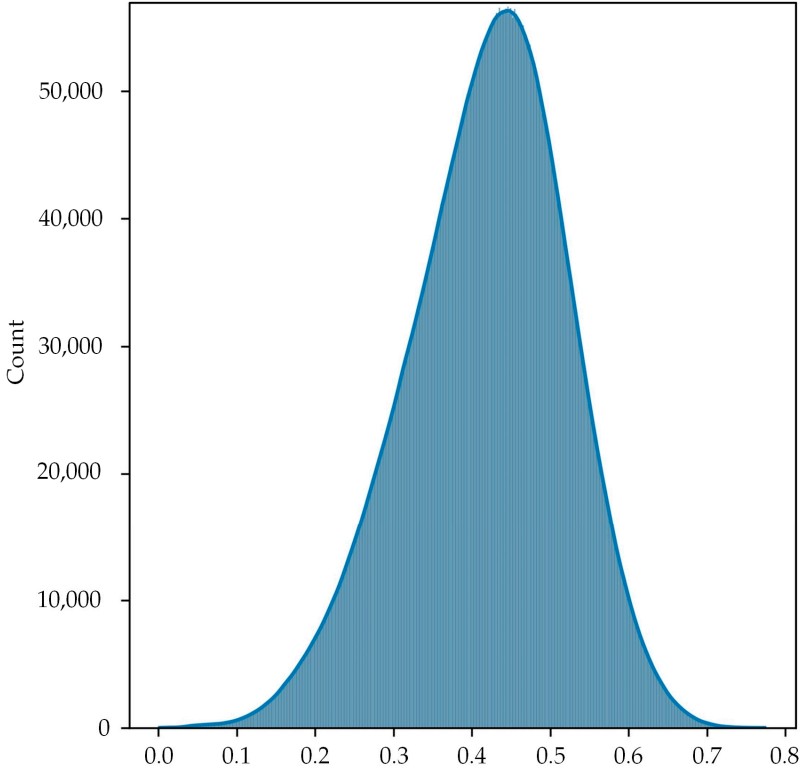

**Figure 7.** Histogram distribution of image cosine distances in the visual map.

*3.3. Optimization*

After obtaining the semantic similarity structure and hash similarity matrix, Formula (7) needed to be continuously optimized. The mini-batch gradient descent (MBGD) method was a better choice, since it combines both batch gradient descent (BGD) and stochastic gradient descent (SGD) methods to accelerate the convergence of the model and considerably reduce the number of iterations required for convergence while ensuring the accuracy of the training results. Nevertheless, it is difficult to define the initial learning rate of this method,

and it falls easily into a local optimum solution. As a result, the learning rate of the MBGD algorithm was further optimized in this work.

Nowadays, the optimization algorithms for adaptive learning rates include Ada-Grad [21], RMSProp [22], and Adam [23]. The AdaGrad algorithm adjusts the learning rate according to the size of the gradient. When the gradient is large, the learning rate will decline significantly. Correspondingly, when the gradient is relatively small, the learning rate will drop less. Although this strategy produces superior results for convex optimization problems, it is not relevant to all deep-learning models. The RMSProp algorithm is a further improvement of the AdaGrad algorithm with an additional decay factor. Compared to the cumulative gradient approach of the AdaGrad algorithm, RMSProp accelerates the convergence by discarding some of the previous values, and the algorithm performs better under non-convex conditions. The Adam algorithm is essentially RMSProp with added momentum, which combines the advantages of the AdaGrad algorithm that make it good at dealing with sparse gradients and the RMSProp algorithm, which is good at dealing with non-stationary targets. Considering that the optimizer may have some influence on the performance of the model, the experiment was conducted on the dataset, the associated mAPs were calculated, and the acquired results are shown in the Figure 8. The experimental findings demonstrate that the mAPs achieved by the Adam optimizer and the AdaGrad optimizer are greater; nevertheless, the AdaGrad curve is unstable with a large amplitude. The reason for this is that the convergence speed of the AdaGrad optimizer for the loss function in this paper is too fast, which causes the descending result to jump repeatedly near the extreme value of the loss function. However, it is difficult to reach the extreme point, thus missing the optimal global solution. Hence, in this paper, Adam was chosen as the method for learning rate optimization in the experiments.

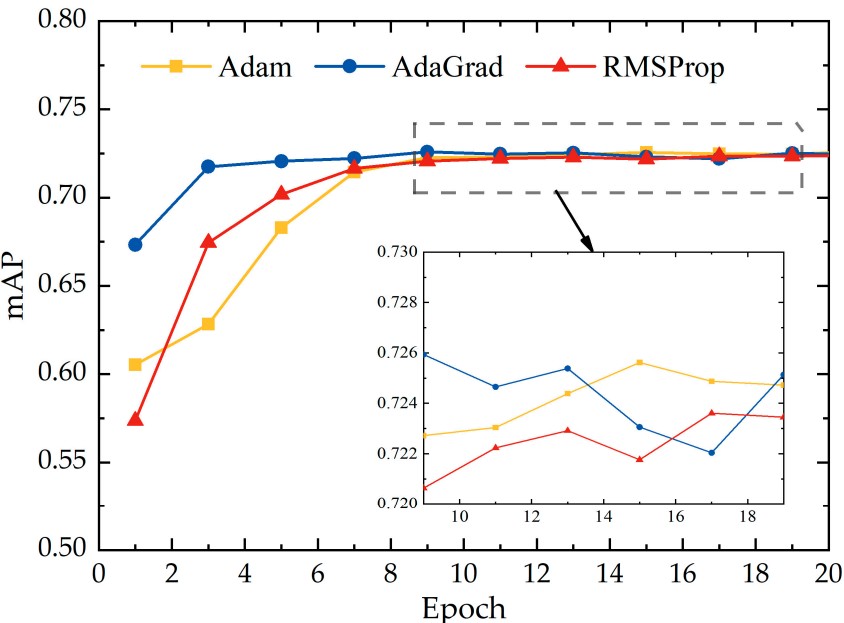

**Figure 8.** mAP of three optimizers.

## 4. Experiment

### 4.1. Database

To verify the feasibility of the model in this paper, FLICKR25K was used as the baseline dataset for comparison with other methods. Moreover, the model proposed in this paper was also applied to the established visual map to observe the model effects.

(1) FLICKR25K contains 25,000 images, and the images are already annotated. The dataset has 24 category labels in total, and each image corresponds to one or more categories, making it a multi-label dataset.

(2) In this paper, the captured images were preprocessed separately for rotation, brightness, and saturation, taking into consideration the different weather conditions, the variations in brightness and darkness of indoor ambient light, and the possible existence of a particular rotation angle when the user takes the image. The dataset for the relevant part of the visual map is shown in Figure 9. Due to the strong structural similarity and repetition of decorations in the indoor scenes in the dataset, a total of 97 categories and corresponding label IDs were set in this paper, and Table 2 illustrates some of the categories and corresponding label IDs in the visual map.

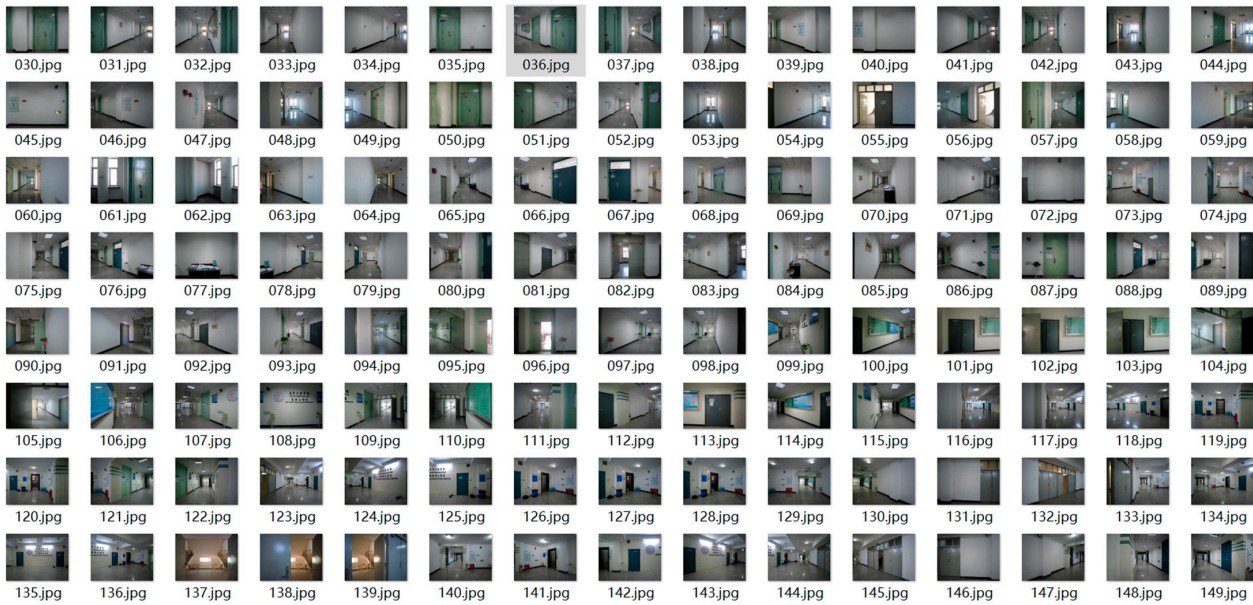

**Figure 9.** Example of the visual map.

**Table 2.** Examples of partial annotations for visual map.

| Label ID | Category | The Number of Images | Label ID | Category | The Number of Images |
|---|---|---|---|---|---|
| 75 | window | 110 | 68 | heating | 63 |
| 74 | table | 31 | 67 | glass door | 20 |
| 73 | storage room | 63 | 66 | garbage can | 102 |
| 72 | stairwell | 114 | 65 | flower stand | 189 |
| 71 | stairs | 4 | 64 | fire cabinet | 191 |
| 70 | open elevator | 3 | 63 | extinguisher | 273 |
| 69 | laboratory | 116 | 62 | elevator | 22 |

*4.2. Experimental Design and Results*

4.2.1. Experimental Design

The GPU and the CPU of the computer used in this paper are NVIDIA GeForce RTX 3090 and 18-core Inter(R) Core (TM) i9-109080XE CPU @ 3.00GHZ, respectively. Considering that $\delta$ is a very critical hyperparameter in the constructed deep hashing model, which seriously affects the performance of the system, an experiment was designed to determine the hyperparameter. Based on the cosine distance histogram, the total probability of semantic similarity was set to 49.8%, 47.7%, and 34.1%, corresponding to hyperparameters 1, 2, and 3, respectively, to explore the effect of these hyperparameters on the retrieval performance of the model. As shown in Table 3, this paper used as an evaluation criterion the mean average precision (mAP), which is one of the primary criteria for evaluating retrieval accuracy.

**Table 3.** mAP with different hyperparameters.

| Hyperparameter | 16 bits | 32 bits | 64 bits | 128 bits |
|:---:|:---:|:---:|:---:|:---:|
| $\delta = 1$ | *0.7324* | *0.7410* | *0.7458* | *0.7482* |
| $\delta = 2$ | 0.7112 | 0.7266 | 0.7386 | 0.7394 |
| $\delta = 3$ | 0.5599 | 0.6293 | 0.6033 | 0.6609 |

4.2.2. Results and Discussion

Observing Table 3, when the hyperparameter $\delta = 1$, the average accuracy is the highest, with better image retrieval capability regardless of whether the code length is 16 bits, 32 bits, 64 bits, or 128 bits. The explanation for this might be that a higher probability of semantic similarity is chosen, allowing the semantics in the picture to be better preserved. In order to verify the accuracy of the proposed algorithm, it was compared with other hashing image retrieval algorithms on the FLICKR25K public dataset. This comparison included three state-of-the-art traditional unsupervised hashing methods, namely ITQ [24], SH [25], and DSH [26], and the deep unsupervised hashing methods SGH [27], DeepBit [28], SSDH, and DistillHash [29] which have been proposed in recent years. The results in Table 4 show that the performance of deep-learning-based algorithms is usually significantly better than that of traditional methods. However, when no proper supervisory signals are found, the powerful representation ability of deep learning cannot be expressed, and its performance may not be as strong as that of traditional unsupervised hashing methods, as in the cases of SGH and DeepBit. In addition, the experimental data also demonstrate that the suggested approach can achieve greater mAP values and more accuracy for a variety of hash code lengths.

**Table 4.** mAP with other unsupervised methods on FLICKR25K.

| Methods | 16 bits | 32 bits | 64 bits | 128 bits |
|:---:|:---:|:---:|:---:|:---:|
| ITQ | 0.6492 | 0.6518 | 0.6546 | 0.6577 |
| SH | 0.6091 | 0.6105 | 0.6033 | 0.6014 |
| DSH | 0.6452 | 0.6547 | 0.6551 | 0.6557 |
| SGH | 0.6362 | 0.6283 | 0.6253 | 0.6206 |
| DeepBit | 0.5934 | 0.5933 | 0.6199 | 0.6349 |
| SSDH | 0.7240 | 0.7276 | 0.7377 | 0.7343 |
| DistillHash | 0.6964 | 0.7056 | 0.7075 | 0.6995 |
| Ours | *0.7324* | *0.7410* | *0.7458* | *0.7482* |

To demonstrate the results of this method, precision–recall curves and top-N-precision curves for this method and the other comparative methods were plotted, as shown in Figure 10. Precision–recall is a good criterion of overall performance and reflects the precision of different recall levels. Top-N precision reveals the average ratio of similar instances in the top-N retrieved instances of all queries, which was calculated by Hamming distance in this paper. Consistent with the mAP results, this paper's proposed method achieved the best top-N-precision results among all methods. Moreover, the precision–recall curves further demonstrate that the proposed method has better hash search results. The above experimental results all prove the superiority of the proposed method.

Experiments will be conducted on the created visual map to validate the model's ability to realize image retrieval in the actual indoor environment and provide significant technical support for indoor visual positioning. Nevertheless, owing to the limitations of the scale of the visual map, this paper added the images from the public dataset FLICKR25K as the interference set. A total of 9900 images from FLICKR25K and 100 images from the visual map were selected as the training set, 1980 images from the FLICKR25K dataset and 20 images from visual map were used as the test set, and 19 labels from the visual map label set, together with the original 24 labels from FLICKR25K, were used as the new label set. Experiments were performed on the recreated dataset, and the mAP was calculated

to obtain the results shown in Figure 11. In addition, Figure 12 shows the precision–recall curves and top-N-precision curves for this paper's method and the SSDH method as used on the new dataset. Experimental results show that the proposed method has higher mAP values than the SSDH method for the constructed dataset. Precision–recall curves and top-N precision also prove that the algorithm in this paper shows better performance on the constructed dataset.

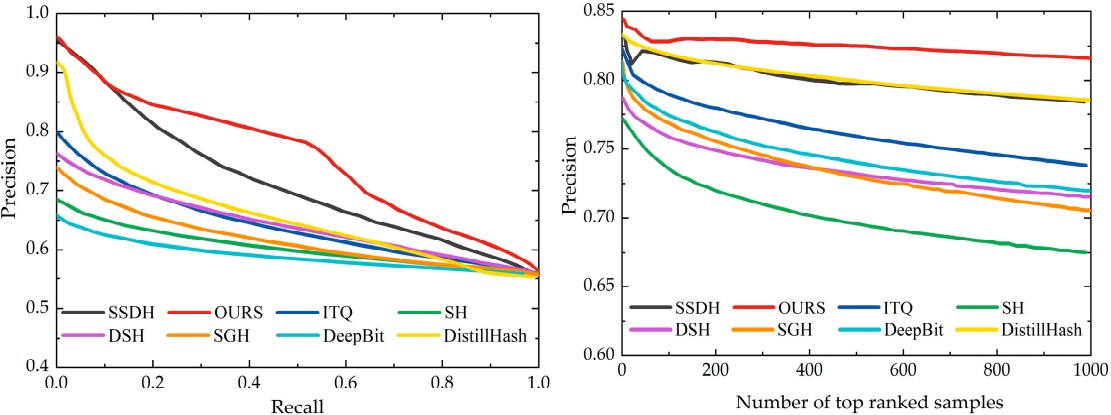

**Figure 10.** Precision–recall curves and top-N precision with code length 32.

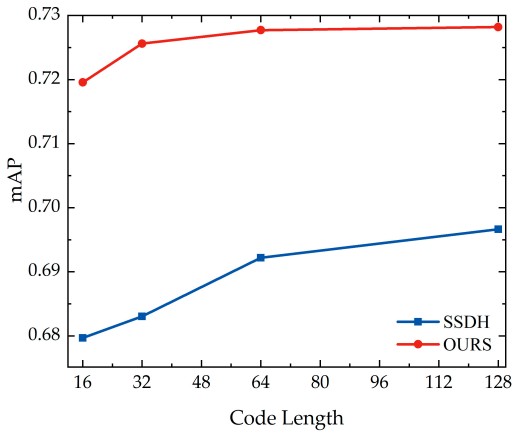

**Figure 11.** mAP on Visual Map.

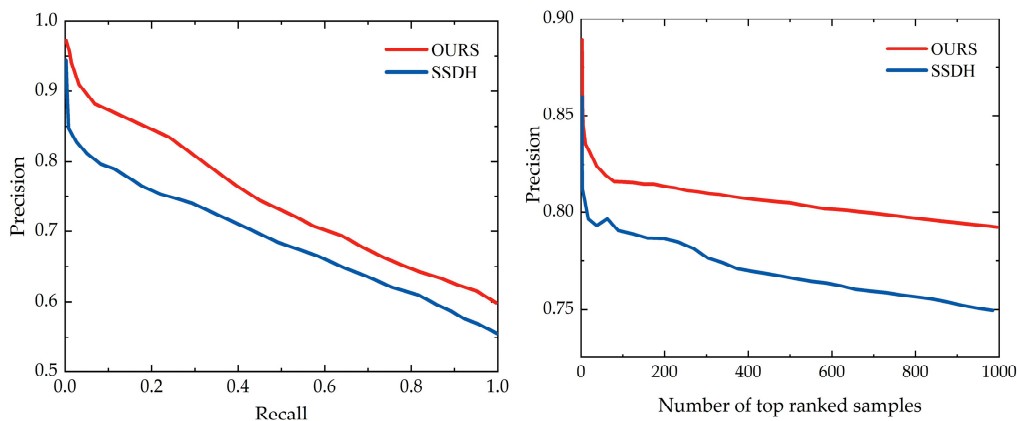

**Figure 12.** Precision–recall curves and top-N precision with code length 32 on visual map.

A further experiment in relation to the time cost of the method considered the impact of image retrieval time on real-time positioning. For code lengths of 16, 32, 64 and 128 bits, the average encoding times of the proposed method and SSDH were as shown in Table 5.

The encoding time was controlled within 0.09 s for all the different code lengths. Under different encoding lengths, the encoding time difference between the improved method and SSDH was not significant.

**Table 5.** Encoding time on visual map.

| Methods | 16 bits | 32 bits | 64 bits | 128 bits |
|---|---|---|---|---|
| SSDH | 0.086673 s | 0.087979 s | 0.087694 s | 0.088878 s |
| Ours | 0.087017 s | 0.089920 s | 0.089114 s | 0.089756 s |

## 5. Conclusions

This paper proposed a visual map construction method based on pre-sampled image features matching, and according to the epipolar geometry of neighbouring location images. The proposed method determined the static sampling interval based on the matching of the feature points between the baseline image and each position image, in a way that effectively balanced the scale of the database and the integrity of the image information. In addition, this paper also proposed a deep hashing-based image retrieval method and designed a loss function based on the characteristics of the log-cosh function curve, namely its being smooth and not susceptible to outliers. The proposed method achieved higher average accuracy in the FLICKR25K baseline dataset, which proved the effectiveness of the method. At the same time, the retrieval results could be returned in sub-second time on the visual map, enabling fast image retrieval with guaranteed accuracy.

**Author Contributions:** Methodology, software and writing—original draft preparation, Jianan Bai; formal analysis, Jianan Bai and Danyang Qin; resources and project administration, Danyang Qin, Jianan Bai, and Ping Zheng; supervision, Danyang Qin and Lin Ma. All authors have read and agreed to the published version of the manuscript.

**Funding:** This work was supported by the Outstanding Youth Program of Natural Science Foundation of Heilongjiang Province (YQ2020F012), the Open Research Fund of the National Mobile Communications Research Laboratory, Southeast University (No. 2023D07), the National Natural Science Foundation of China (61771186), and the Postgraduate Innovative Science Research Project of Heilongjiang University in 2022 (YJSCX2022-205HLJU, YJSCX2022-080HLJU).

**Data Availability Statement:** The data presented in this study are available on request from the corresponding author. The data are not publicly available due to privacy considerations.

**Conflicts of Interest:** The authors declare no conflict of interest.

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
