# Peer review of "Image Retrieval Method Based on Visual Map Pre-Sampling Construction in Indoor Positioning"

_ijgi, doi:10.3390/ijgi12040169_

Round 1
Reviewer 1 Report
The methods proposed by the authors have some novelty, and there are several suggestions:
1. The contents between lines 63-65 and 101-103 are partially duplicated.
2. It is recommended to compare the generalization performance of the proposed methods.
3. It is suggested to add the actual positioning coordinates, and if the indoor environment changes locally, can the proposed method still be used?
Author Response
Thanks for your earnest review of our work, and we have made corresponding revisions and responses to your valuable suggestions. Please refer to the attachment for details.

Reviewer 2 Report
This paper proposed a visual map construction method based on features matching in order to balance the scale of the database and the integrity of the image information. Experiments on the FLICKR25K dataset and the Visual Map proved that the method proposed in this paper could achieve sub-second image retrieval under the premise of ensuring accuracy and showing its promising performance. However, there are some issues in the manuscript that need to be addressed, some of them are listed below.
[1] The clarity of all the figures in the paper is very poor, such as Figures 1 to 4, etc., the clarity of these figures cannot meet the requirements for the publication of the paper, and it is even difficult to see the key instructions in the figures clearly, such as Figure 6, etc. It is suggested to improve the clarity of all figures.
[2] Because the experimental environment is primarily adorned with walls and doors, and the structured information is excessively repetitive, so the authors suggest that the coverage and matching ratios are proposed to be 0.7 and 0.5, respectively. It is suggested that the authors further analyze the optimal coverage and matching ratios, such as through mathematical models, rather than simply judge the two key values according to the experimental environment.
[3] The proposed method determined the static sampling interval based on the matching of the feature points between the baseline image and each position image, which could effectively balance the scale of the database and the integrity of the image information. In the line 203, the authors describe that the sampling spacing is decided to be 3m*1.8m due to the narrow and lengthy features of the experimental environment. The authors directly give a value of sampling interval, so what makes me confused is how to balance the size of the database and the integrity of image information.
[4] From lines 294 to 307, the authors describe three algorithms, AdaGrad, RMSProp, and Adam, and finally selects Adam as the learning optimization method. First, it is suggested that the authors provide relevant references when introducing the three algorithms. Secondly, for problem (7), in order to avoid the MBGD algorithm falling into the local optimal solution, what is the sacrifice cost of the author's choice of Adam algorithm.
[5] What is the Mobile MATLAB.
Author Response
Thanks for your review of our work, and we have made corresponding revisions and reply to your valuable suggestions. Please see the attachment for details.

Reviewer 3 Report
This paper presents a deep hashing based image retrieval method in the visual indoor positioning. The topic is interesting and the results seems promising, but the structure of the paper should be improved. There are many duplicated information or useless data description, which should be removed. In addition, the English writing should be extensively improved.
1. The title of this paper should be more specified. Current title is a very general topic which cannot reflect your contribution or characteristics of your algorithm.
2. Figure 1 contains few useful information. It is difficult to understand the ‘framework’ of the algorithm via this figure. Reconsider it.
3. What is the difference between the proposed deep harsh algorithm and the method in reference [20]? Why your algorithm outperforms the SSDH algorithm?
4. All figures in the manuscript are too blur. Replace them with higher resolution ones.
5. There are some duplicate sentences in the introduction section and related work section, which can be trimmed.
6. In table 4, the reference algorithms should be briefly introduced, so that it makes the comparison clearer.
7. The results were not analyzed and interpreted.
8. What is the relationship between data described in section 3.1 and FLICKR25K? Is your own data used for algorithm evaluation?
Author Response
Thanks for your review of our work, and we have made revisions and responses to your valuable suggestions. Please refer to the attachment for details.

Round 2
Reviewer 1 Report
The authors can do further research on practicality.
Reviewer 2 Report
The authors have answered my comments, acceptance is suggested
Reviewer 3 Report
Thanks for the author's response. All the concerns have been addressed and I don't have new conment now